

# Documentary evidence of urban droughts and their impact in the eastern Netherlands: the cases of Deventer and Zutphen, 1500– 1795

Dániel Johannes Moerman[1]

[1]Faculty of Humanities, Department of Art and Culture, History, Antiquity, Vrije Universiteit Amsterdam, De Boelelaan 1105, 1081 HV, Amsterdam

*Correspondence to:* Dániel Moerman d.j.moerman@vu.nl

**Abstract**: Compared to other parts of Europe, very little is known about pre-instrumental drought periods in the Netherlands. Existing reconstructions are based primarily on data from England, France, and Germany, while more precise, local studies on drought and its impact are still absent. This article thus aims to further our knowledge of droughts in the Netherlands between 1500 and 1795, by focusing specifically on drought in an urban context to provide a more precise and local idea of the impact and severity of drought. The main case studies are cities in the eastern part of the country, Deventer and Zutphen. Both cities lay in relative close proximity to each other and share similar geological and hydrological conditions, as well as extensive archives that can be used to gather documentary data regarding historical drought periods. The three primary aims of the article are: 1) to examine the potential use of documentary data from the city archives of Deventer and Zutphen for historical drought reconstruction; 2) to establish droughts for both cities on the basis of the year, month/season in which they took place, as well as ranking the droughts according to the impact-based Historical Severity Drought Scale (HSDS) and 3) to compare the data from this analysis with that of other indices. In the end, the article strengthens the need to focus on documentary data from local case studies regarding drought, not only to provide more precise local reconstructions of drought-severity compared to regional studies, but also to take into account the long-term effects on urban waterscapes and the provisioning of fresh water.

## 1. Introduction

In recent years, droughts have become a more pressing topic of research. Worldwide, droughts of varying severity affect societies, whether on an agricultural, hydrological, or on wider socio-economic level, which is expected to increase within the current trends of climatic change (Kchouk et. al., 2021; Savelli et. al., 2022; Spinoni et. al., 2018). The study of past droughts for the pre-instrumental period on the basis of documentary evidence and natural proxies, such as dendroclimatology, has displayed the possibility to reconstruct drought-events and their societal impact in Europe, which has led to the development of several historical drought reconstructions and indices. (Bauch et. al., 2020; Brázdil et. al., 2016/2018/2019/2020; Camenisch et. al., 2020; Garnier , 2019; Kiss, 2017/2020; Leijonhufvud and Retsö, 2021; Piervitali and Colacino, 2001; Pribyl and Cornes, 2020; Stangl and Foelsche, 2022). However, very little to no historical drought data exists for the Netherlands. The limited data available from the voluminous works of Buisman (1995/1996/1998/2000/2006/2015) is based primarily on reconstructions and sources from England, France and Germany, and sporadic sources from across the Netherlands. A recent study by Camenisch and Salvisberg (2020), has emphasised the need to analyse regional



and local aspects of droughts by studying geographically limited source samples, such as municipal data from city
archives. Compared with other, supra-regional drought indices, this can lead to a more detailed understanding of
the extent and severity of certain droughts on a local level, while also providing insights into previously unknown
droughts. Even droughts with a larger geographical footprint, such as the infamous 1540 'Megadrought' (Wetter
et. al. 2014), can thus demonstrate a greater temporal diversity if more localised data is included in the analysis
(Maughan et. al. 2022). As such, the data provided by Buisman cannot suffice to study the local or regional severity
and impact of drought for the Netherlands, and, as follows, further research is needed.
This article aims to further our knowledge of pre-instrumental droughts in the Netherlands between 1500 and 1795,
focusing on two cities in the eastern part of the country – Deventer and Zutphen. Both have rich municipal archives,
relatively similar geological and hydrological conditions, and lay in close proximity to one another. The eastern
Netherlands,  are well-known as a region more prone to precipitation deficits compared to the coastal regions in
the west, making it more susceptible to drought. This is a trend that likely played a role in the past and is estimated
to cause increase drought-risks in particularly the eastern regions, which makes the emphasis on this region of past
as well as future value (Phillip et. al., 2020).
The focus on more specific urban contexts also moves away from the focus on agricultural drought, which is
dominant in historiography, shifting the emphasis to the wider hydrological and socio-economic impact of drought
within a city's walls. Common factors to denominate drought severity according to the Palmer Drought Severity
Index, or PDSI (Palmer, 1965), such as temperature, precipitation levels and soil-moisture deficits, are not enough
to determine the impact of droughts on urban environments. Urbanisation, and other large-scale influences of
human actions on the distribution and use of water, have often been ignored in many classical drought indices that
focused primarily on precipitation and temperature data (Briffa, Van Der Schrier and Jones, 2009; Savelli et. all.,
2022). Many previous studies into past droughts worked in relative isolation, without taking into account the
complex interactions between natural and human processes in the hydrological sphere (AghaKouchak et. al., 2021;
Van Loon et. al. 2016; Maughan et. al. 2022; Mukherjee, Mishra and Trenberth, 2018; Vörösmarty et. al., 2004)).
These factors are already more present in another index, the so-called Historical Severity Drought Scale (HSDS),
which allows for a reconstruction of drought based on a systemic inventory of the different hydrological and socio-
economic impacts that constitute levels of drought severity (Garnier, 2014/2019; Metger and Jacob Rousseau,
2020). Looking at urban documentary data thus not only provide more precise local reconstructions of drought-
severity, but can also take into account the long-term effects on urban waterscapes and the provisioning of fresh
water.
This article has three primary aims: 1) examining the potential use of documentary data from the city archives of
Deventer and Zutphen for historical drought reconstruction; 2) to establish droughts for both cities on the basis of
the year, month/season in which they took place, as well as ranking the droughts according to the impact-based
Historical Drought Severity Scale; and 3) to compare the data from this analysis with that of other indices, such as
the Buisman and IJnsen temperature series for the Netherlands, the supra-regional drought index, or SDI, that
comprises data from Switzerland, France, the Netherlands and Germany, (Camenisch and Salvisberg, 2020), and
the Old World Drought Atlas (OWDA) that provides an overview of dendrochronological drought data on a
regional scale (Cook et.al., 2015).



The article is divided in six sections. The first section provides a detailed overview of the sources used in the
reconstruction of drought for Deventer and Zutphen. Section two will present outcomes from the study of these
sources, by which the drought years are presented via a chronological HSDS. Section three discusses a specific set
of examples from the sources, providing a more detailed analysis of the data and their respective values. Sections
four, five, and six compare the data gathered in this study with other indices, followed by a final discussion and
conclusion.

**2. The data**
To reconstruct past weather and climatic phenomena, historical climatologists draw from a large amount of
documentary sources that provide either direct or indirect (proxy) data about changes in weather or abnormal
patterns of precipitation and temperatures (Brázdil et. al., 2005/2010; Pfister, 2018). For drought reconstructions,
the commonly used documentary evidence consists of annals, chronicles, and diaries, in which people recorded
daily or extraordinary weather situations, or more institutional sources, such as tax and harvest records, and
religious data with regard to rogation ceremonies (Brazdil et. al. 2013/2019/2020; Dominguez-Castro et. al., 2012;
Kiss and Nicolic, 2015). Municipal records, from towns or villages, become more systematised from the end of
the fifteenth century onward, often containing deliberations and resolutions that indicate means by which local or
state governments aimed to alleviate the effects of drought or other weather extremes (Garnier, 2019; Gorostiza,
Escayol and Barriendos, 2021; Grau Satorras et. al., 2021). Therefore, municipal archives qualify as a good
*Fundgrube* for (proxy) evidence of past droughts.
For this study, the municipal archives of two cities in the eastern Netherlands, Deventer and Zutphen, have been
studied extensively in search of references to drought-related phenomena. Deventer and Zutphen are both situated
along the IJssel river on sandy river dunes from the Holocene and relied on surface water from the rivers and clean
groundwater for everyday use (Vogelzang, 1956). The primary sources that have been studied were primarily
official municipal records, such as daily resolutions from the city government, ordinance books, and petitions. For
Deventer, a long-running series of sources was available in the form of the so-called '*Edicta magistratus*' and
'*Liber publicationum*', which consist of books running continuously from 1459 until 1795, listing chronological
ad-hoc resolutions and ordinances taken by the magistrates to cope with problems threatening public safety and
welfare on a short notice. These were complimented by the '*Protocollen des Rhades*', or the general daily
resolutions, which were available from 1566 until 1795, as well as the books of 'concordances' from the middle
of the sixteenth to the late eighteenth century, which contains petitions from the collective of neighbourhood
representatives known as the 'sworn men' to the magistracy. For both the daily resolutions and books of
concordances, alphabetical reference books from eighteenth and nineteenth-century authors are also available that
provided a useful, yet also limited tool to find certain relevant entries regarding drought. In the case of Zutphen,
the extensive series of daily resolutions and can be studied from 1573 until the start of the nineteenth century. This
series, including the very detailed and digitised reference books provided the primary source for Zutphen.
In order to identify periods of drought, an extensive and serial study of the above-mentioned sources was required.
Where available, the reference books were used as additional tools to find entries connected to drought-related
issues, such as water provisioning, fire, watermills, and other matters related to waterworks and shipping, as well





as a dearth in foodstuffs and other items as a result of drought. Firstly, the sources for Deventer were studied,
beginning with the *'Edicta magistratus'* and *'Liber publicationum'*, which were studied on a year-by-year basis in
which all entries were searched for direct or indirect references to drought. This yielded many results that formed
the basis of the following archival research. Second in line were the books of concordances, which were also
studied on a year-by-year basis. The daily resolutions were not studied on a year-by-year basis because of the
density of the information recorded in these books it would simply be too time-consuming. Instead, the daily
resolutions were studied only on the basis of the reference books and the findings from the '*Edicta magistratus'*
and *'Liber publicationum'*. In this case, not only the drought years found in the previous sources were searched in
the daily resolutions, but also two years before and after, given the insidious nature of drought and possibility that
source might display certain developments of a drought on an earlier basis. After the study for Deventer was
completed, the study of Zutphen started off with an analysis of the largely digitised reference works for the daily
resolutions. The earlier discovered drought years for Deventer were also used as reference points, and were used
to study specific years, including the years before and after, in the daily resolutions.
For both cities, the rough data was first copied into separate databases for each city, after which the data were
filtered by setting aside references that did not directly relate to drought. These included references to future
measures to be taken when severe droughts would occur, or measures where the relation to drought was less clear.
Secondly, the remaining drought-events were filtered for each city according to drought-type (meteorological,
agricultural, hydrological, socio-economic) and season. Hereafter a chronological database was created combining
the data from Deventer and Zutphen as a chronological overview of the specific drought events for each year. This
specific overview was also used for the next step: ranking the severity of each drought per year.

**3. Methodology**
In this section I discuss several indices and explain the particular choice for the HSDS as the preferred method to
rank the severity of the droughts for Deventer and Zutphen. Many historical drought reconstructions have been
done on the basis of natural proxy-data from dendroclimatological reconstructions. These focus on tree-ring
analysis to reconstruct tree growth that provides insights into precipitation and temperature levels. This can be
expressed along the PDSI as an estimate of relative dryness based on reconstructions of temperature and
precipitation (Brázdil et. al. 2018). Certain long-term dendroclimatological reconstructions, such as the OWDA
for Europe and parts of North-Africa use a self-calibrating PDSI (scPDSI) to create year-by-year maps of
reconstructed summer droughts on a 5414-point half-degree longitude-by-latitude grid. The scPDSI has a high
degree of spatial comparability across a broad range of climatological regions, which allows for comparisons with
other pre-instrumental droughts, for example in North-America (Cook et. al. 2015).
One of the most commonly used indices to categorise drought-severity in Europe is based on the seven-point
ordinal index devised by Pfister during the 1980s, also named 'Pfister Indices' (Brázdil 2020; Nash et. al., 2021;
Pfister, Camenisch and Dobrovolný, 2018). These indices can indicate both temperature differences and variations
in precipitation. In the seven-point index for precipitation, values ranging from rather wet to extremely wet (+1 to
+3) and rather dry to extremely dry (-1 to -3) are used to typify periods on the basis of direct or proxy-based
information regarding precipitation within a certain area. Such an index cannot be built on descriptive documentary



evidence alone but should also include proxy-data, such as evidence from plant-phenology and dendroclimatological analysis. A merely descriptive index would only be able to use a three-point scale, only taking into account the extraordinary (-1 or +1) as a deviation from the average (0). Every seven-point index also needs a reference period to denote the deviations from the average, which often consists of a series of instrumental measurements from the period prior to the full onset of global warming, most commonly 1906 to 1960 (Pfister, Camenisch and Dobrovolný, 2018).

Several studies into historical droughts within Europe have applied the seven-point index as a means to indicate the severity of past droughts (Bauch et. al., 2020; Brázdil et. al. 2013; Camenisch and Salvisberg, 2020; Leijonhufvud and Retsö, 2021). However, there are also certain limits to the seven-point index. Kiss and Nikolić (2015), for example, remarked that the requirements for the index can hardly be met for the European Middle Ages, where the amount of available documentary evidence is often insufficient to estimate the severity of drought on a month-by-month basis. In their attempt to create a 400-year long drought-index for the cities of Bern and Rouen, Camenisch and Salvisberg (2020) similarly argue that, given the available data from both cities – primarily chronicles and municipal records from the fourteenth to the early eighteenth century – did not allow for all three index values (-1 to -3) to be used. The sources from both city's only provide instances of extreme drought events, which left a significant mark on inhabitant's memory and prompted city governments to take action. Therefore, instead of using all three values, only extremely dry (-3) and very dry (-2) were used in their analysis, considering that the more frequent and less impactful droughts (-1) were usually not recorded. For both cities, most droughts during the 400-year period were characterised as very dry (-2), and only a few instances were classified as extremely dry (-3). The survey also led to the identification of specific accumulations of droughts, for instance, at the end of the fourteenth, second half of the sixteenth, and the 1670s and early, as seasonal difference was discovered as the droughts in Bern often took place during the summer, while those in Rouen were more prevalent in spring.

The previous conclusions can also be applied for the corpus of municipal sources that have been investigated for Deventer and Zutphen. However, the documentary data from Deventer and Zutphen does not allow for a precise month-by-month reconstruction, as the duration of the droughts is not mentioned in the primarily descriptive data. To categorise such droughts into a seven-point index, monthly records of precipitation are required. In this case, a drought can only be denoted as 'very dry' (-2) after at least a one-and-a-half months of reduced precipitation, while the value of 'extremely dry' (-3) is reserved for two or more months without rainfall (Camenisch and Salvisberg, 2020). As the data from both Deventer and Zutphen do not give exact insights into the length of certain droughts, only referring to 'long' or 'prolonged' periods of drought, which do not indicate a specific timeframe, the seven-point index cannot be applied. However, the primary references to drought concern descriptions of its human and economic impact on a societal level, which are also more accurate representations of past perceptions of drought than modern conceptions of precipitation and evaporation (Garnier 2015). This data can be used according to the HSDS to delineate droughts on an impact-centred scale. The HSDS distinguishes droughts on the basis of societal reactions that can be found in various sources, which are classified in categories on a 1 to 5 scale (see table 1) from an absence of precipitation to full-scale social crisis. An additional category is -1, which denotes instances where both qualitative and quantitative data are considered insufficient, while a drought reference is kept solely for chronological reconstruction (Garnier, 2014).





*Table 1: Historical Severity Drought Scale (for the sixteenth to nineteenth centuries), from Garnier (2014)*

| Index | Description |
|---|---|
| 5 | exceptional drought: no possible supply, shortage, sanitary problems, very high prices of wheat, forest fires |
| 4 | severe low-water mark: navigation impossible, lay-off of wheatmills, search for new springs, forest fires, death of cattle |
| 3 | general low-water (difficulties for navigation) and water reserves |
| 2 | local low-water in rivers, first effects on vegetation |
| 1 | absence of rainfall: rogations, evidences in texts |
| -1 | insufficient qualitative and quantitative information but the event is kept in the chronological reconstruction |



**4. Outcomes**

Based on the indicators of drought and its severity in the studied sources, an HSDS index has been constructed
including both data from Deventer and Zutphen (see fig. 1). The index ranks droughts on an annual basis using the
five-point scale, although instances of purely meteorological droughts (scale 1) and its effects (rogation ceremonies
and public prayer) have not been discovered. In total, 33 years with drought have been reconstructed, 26 for
Deventer, 16 for Zutphen, and only nine coinciding years. Hydrological droughts with a significant impact on the
city's waterway's and the availability of water (scale 3) are amongst the most common forms of drought described
in the sources, occurring 24 times. More extreme hydrological conditions, those within scale 4, are less common
but still make up a significant part of the recorded droughts, namely nine instances. Scale 5, denoting full-scale
societal crisis and critical shortages of food and water, has not been identified.

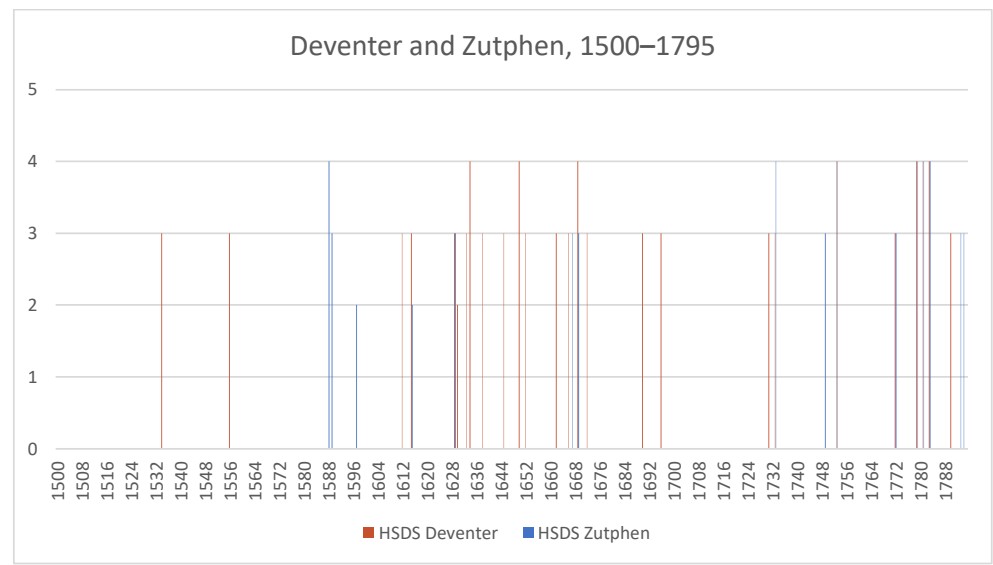

**Figure 1: Chronology and severity levels of droughts within Deventer and Zutphen according to the Historical Severity Drought Scale (HSDS), 1500–1795.**



The most common types of drought which are mentioned in documentary sources refer to instances of meteorological drought, describing a deficiency of precipitation, agricultural drought, which describes the effect of meteorological drought on agricultural production, hydrological drought, which relates to a shortage of water in watercourses, lakes and underground water tables, and socio-economic drought, when the effects of drought cause widespread economic and societal disruption, most commonly in the form of subsistence crises (Brázdil et. al., 2018; Wilhite and Pulwarty, 2017). As municipal records usually only contain references to extreme weather events, the descriptions of drought in the sources refer almost exclusively to extremities (Camenisch and Salvisberg, 2020; Garnier, 2019). With regard to both Deventer and Zutphen (see fig. 2), hydrological drought is by far the most common type of drought described in the sources. In most cases, this refers to low water levels or a complete lack of water in certain rivers and canals, as well as a shortage of water in wells and pumps. Meteorological drought is more prevalent in sources from Deventer, although in general the descriptions refer exclusively to 'excessive', 'strong', 'prolonged', or 'long-lasting' periods of drought, often accompanied with a reference to the hydrological effects, such as dried up waterways and wells. Agricultural drought occurs very rarely in the sources, as there is only one reference from Deventer that explicitly mentions negative agricultural yields as a result of a severe drought. Last but not least, socio-economic drought only occurs during very strong droughts, usually the result of an accumulation of events leading to a severe lack of water and a shortage of food and other goods.





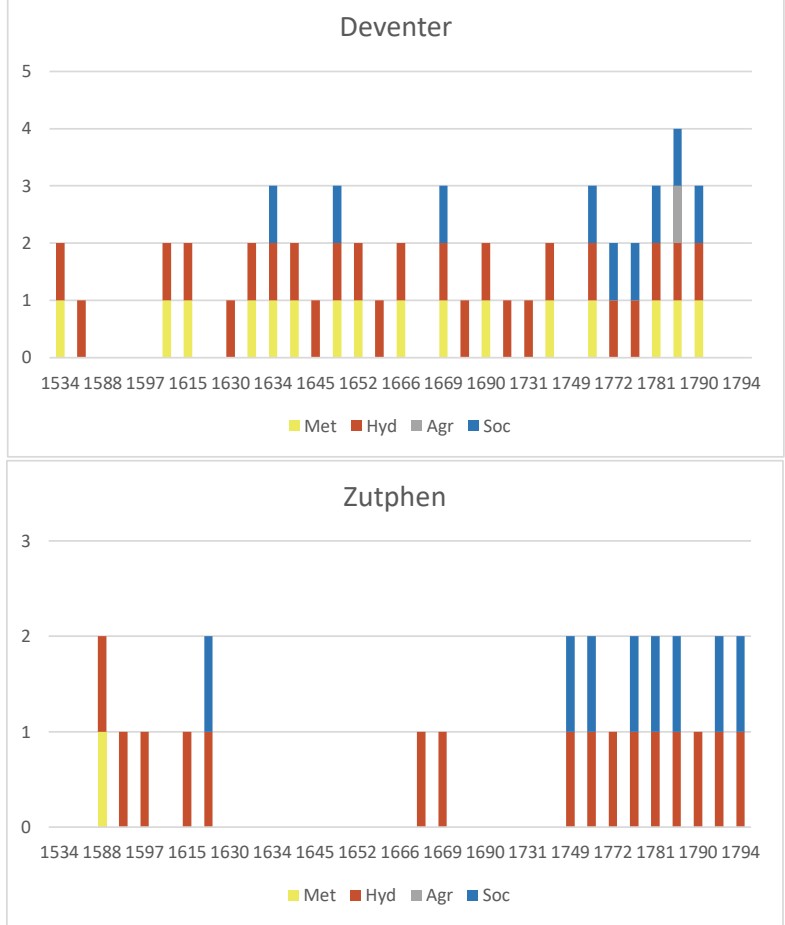

**Figure 2: Difference in drought types, meteorological (Met),hydrological (Hyd), agricultural (Agr) and Socio-economic (Soc), according to each year for Deventer and Zutphen, 1500-1795**

While there are a number of different drought years for both cities, there are specific years that coincide, although not always in terms of severity. The year 1615, for example, is ranked 3 for Deventer, yet 2 for Zutphen. The sources for Deventer for 1615 indicate both a period of drought and lack of water, while Zutphen did not seem to suffer from the low water levels on the IJssel river. However, most coinciding years, such as 1733, 1753, 1772, 1779, 1781, and 1783, indicate similar levels of drought severity for both cities in terms of seasonality.

A notable level of difference between the two cities is that of seasonality (see fig. 3). Deventer seems to have a much higher rate of spring  droughts – recorded between March and May – and summer droughts – recorded between June and August –, while Zutphen displays a larger amount of winter droughts – recorded between December and February. Both cities seem to have witnessed an equal amount of autumn droughts – recorded between September and November –, which, together with summer droughts constitute the most common category of droughts based on seasonality.



Similar to the research by Camenisch and Salvisberg, the results for Deventer and Zutphen also display specific
clusters or accumulations of drought years that took place within a span of several, sometimes subsequent years.
Droughts with a moderate to severe impact, ranking 3 or 4 on the HSDS, occurred during the years 1630–1640,
1650–1652, 1662–1669, 1731–1733, 1781–1783, and 1790–1794. This does not include years in which references
are made to the damaging effects of previous droughts, often a year or even multiple years after a severe drought
occurred. Most of the severe droughts ranking 4 on the HSDS occurred during the second half of the eighteenth
century, between 1753 and 1783.

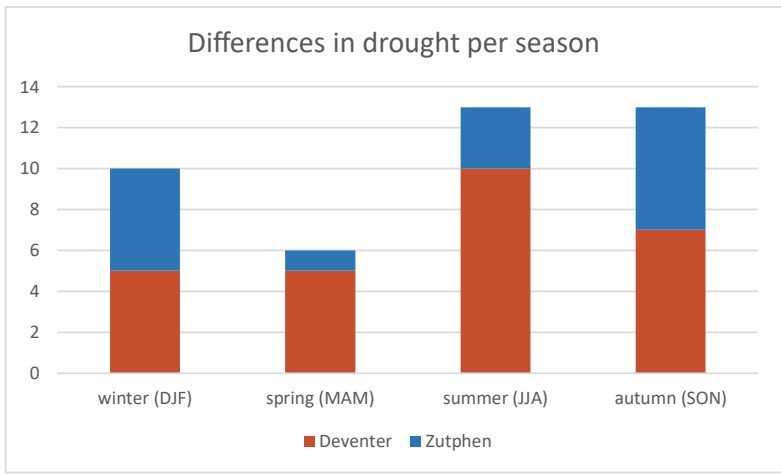


**Figure 3: The number of droughts according to season for Deventer and Zutphen, 1500-1795.**

**5. Examples from the sources**
It would go beyond the scope of this article to dive into the details of each specific drought year discovered for
both cities. A brief overview of these can be found in appendix 1 at the end of the article. Nevertheless, to make
sense of the otherwise rather abstract notions mentioned in the HSDS, it is necessary to provide a number of
detailed examples. The number of examples has been restricted the most extreme and detailed examples, some of
which coincide for both Deventer and Zutphen. These are 1669, 1733, 1753, 1781, and 1783.
**5.1. 1669**
Deventer witnessed a period of severe drought in September 1669, which, according to contemporary records from
the city, led to extraordinarily low water levels on the IJssel river. As a result, many of the wells and pumps the
city were rendered dry and unusable. The citizens and inhabitants suffered from this inconvenience and public
clamour regarding the scarcity of water was heard throughout the city. One of the main concerns, however, was
the risk for fires that could turn the city into a ruin as contemporaries feared. For Zutphen, references to the shortage
of water are less explicit for September that year. Here, no explicit mention of water scarcity is made in the city
governments documentation, but the fear of fire becomes apparent in a resolution that directed the city crier to call
upon all inhabitants to store water in case of an uneventful fire. While the impact of the drought is very explicit



264 for Deventer (scale 4), the reference to compulsory storing of water for Zutphen (scale 3) also implicitly links to

265 hydrological drought but less to a direct societal impact or near-crisis situation.

**5.2. 1733**

267 The year 1733 seems to show the opposite in terms of references. For Deventer, the impact of the drought was felt

268 primarily during the summer, which led to a lack of water in the Schipbeek river that supplied water to the city's

269 harbour and canals. Whether this had an impact on the water levels in the city's wells and pumps, however, is not

270 mentioned. In Zutphen, the 1733 drought was first mentioned in October, when a genever distillery petitioned to

271 the city government that their capacity to produce suffered due to the great shortage of water within the city. In

272 this case, the effects of the hydrological drought are more explicit for Zutphen (rank 4) than for Deventer (rank 3).

273 Nevertheless, it can be assumed that the lack of water in the Schipbeek hampered navigation and the supply of

274 water power to Deventer's watermills.

**5.3. 1753**

276 For the year 1753, equally severe droughts are mentioned for both Deventer and Zutphen in terms of impact. In

277 Deventer, the effects of drought were first felt in June, when an 'excessive drought' (*excessive droogte*) led to a

278 shortage of water in the city's wells. This lack of water led to a general shortage of water that prompted the city

279 government to take action. In Zutphen, the impact of the drought was reported in September, which mentioned the

280 low water levels on both the IJssel and Berkel rivers that led to the 'paralysis' (*verlamminge*) of most wells and

281 pumps. This displays a similarity in drought severity (rank 4), which refers to societal setbacks, for example by

282 limiting water use, rather than a full socio-economic crisis, although the potential for the latter could have been

283 present.

**5.4 1781**

285 For 1781, the severity of drought is indexed equally on the HSDS for both cities (rank 4). In July that year, the

286 Schipbeek was reported to have once again 'consumed' (*verteert*) of water to the detriment of the city, although

287 no further details of the negative impacts are recorded. It can be assumed, however, that the drying up of the

288 Schipbeek must have been felt, as it would have certainly paralysed the watermills. The impact of drought in

289 Zutphen was already felt in February, implying that the drought started in the winter. Here, the drought and low

290 water levels resulted in a lack of navigation via the Berkel river and a limited operation of the city's watermills.

291 However, no effects on the availability of water in both cities' wells and pumps is mentioned.

**5.5 1783**

293 The most detailed drought year (rank 4) recorded for both cities occurred in 1783. In Deventer, the strong and

294 excessive drought led to a lack of water in most of the wells during around the beginning of August. Later during

295 that month, a rare instance of agricultural drought is also mentioned as the a great spring drought led to a reduced

296 yield in buckwheat. This implies that the prolonged drought probably set in during the spring-months, while its

297 effects did not become detriment until the end of the summer when the prices of cereals increased significantly. In

298 Zutphen, the effects were primarily felt by the drying up of the Berkel river, which led to a standstill of all

299 watermills at the beginning of August. Another likely effect of the drought of 1783 was an epidemic of dysentery

300 in both Zutphen and Deventer. In Zutphen, the onset of the epidemic in towns and villages around the city was



noticed in early August, while the first case within the city walls was recorded on the fourth day of that month.
The disease spread rapidly during the following months, and the epidemic must have lasted until the end of
October. The city government in Deventer was aware of the outbreaks of dysentery in surrounding cities at the
time, but the first cases were not reported within the city walls until the beginning of October. Although
contemporary sources suggest no direct link between the lack of clean water and the outbreak of dysentery – which
would have not followed the medical logic of the time – many recent studies suggest that extreme droughts were
likely the main drivers behind some of Europe's largest dysentery epidemics (Brázdil et. al. 2020; Camenisch et.
al. 2020; Garnier, 2019; Pribyl, 2020). In all these cases, the cause of widespread dysentery is attributed by
historians to a lack of clean, fresh water as a result of drought, which prompted people to use polluted water, or to
seek water from unsafe sources.
In general, the source material often refers to similar indicators of hydrological drought, often hindering socio-
economic life, but rarely causing widespread disruption or crisis. Instances of agricultural drought and its effects
on food prices or general subsistence are very rare and only account for one particular case; the year 1783, when
the prolonged drought led to a shortage of water, shutdown of watermills, dearth in cereals, and an outbreak of
dysentery in both cities. However, the sources from that year do not suggest that the situation led to a crisis
situation. There were also notable differences in the responses to drought, which do not correspond one-on-one
for both cities during most years, despite the relative proximity and similarity of both cities in terms of geological
and hydrological circumstances and the systems of water provisioning.

**6. Comparison with Buisman-IJnsen**
Compared to other countries, very little concrete data with regard to temperature and/or precipitation exist for the
Netherlands prior to the instrumental period after 1850. The Royal Netherlands Meteorological Institute (KNMI),
founded in 1854, has a collection of 'antique data', consisting of early instrumental observations from the
eighteenth and early nineteenth century. These datasets are comprised of observations from several weather
stations across the Netherlands. Most of the stations from which eighteenth century records exist are located in the
province of Holland – such as Amsterdam, Alkmaar, Bergen (North-Holland), Delft, Haarlem, Leiden Rijnsburg,
and Zwanenburg – leading to rather regional measurements more typical for the precipitation-rich western
provinces along the North Sea coast, not the inland provinces that are more susceptible to strong droughts. The
early records for the eighteenth century also contain very few consistent records regarding precipitation (Geurts
and Van Engelen, 1992). Most data known for the pre-instrumental period consists primarily of reconstructions
regarding winter and summer temperatures.
The longest list of pre-instrumental, and partially instrumental, estimations of winter and summer temperatures
was compiled by Buismand and IJnsen. Despite its incredible length, running from the year 751 CE until 2000,
this data is generally not well-known outside of Dutch-speaking academia (Van Engelen, Buisman and IJnsen,
2001; Pfister, Camenisch and Dobrovolný, 2018 ). This data-series was constructed with the use of various proxy-
data from the early modern period, such as the weather diary of German pastor David Fabricius for the larger
Frisian area in the north of the Netherlands, a set of frost-day notes from the German city of Kassel, the 'tow barge'
records from De Vries and the Manley (1974) records of monthly temperatures in central England. Buisman and
IJnsen also included data from the aforementioned records of the aforementioned weather stations (1706-1905).





The winter – from November to March – and summer – from May to September – temperatures in this series have
been categorised along an annual nine-point scale from 1 (extremely soft/cool) to 9 (extremely harsh/warm)
(IJnsen, 2010).
For the comparison, only values from 7/-7 to 9/-9, implying above average summer and winter temperatures have
been taken into account as relevant for possible correspondence between drought and above or below average
temperatures. Overall, the result of the comparison was rather meagre. Only a handful of years displayed a
correspondence between cases of moderate to strong and extremely strong droughts – those ranking 3, 4 or 5 on
the HSDS – and above or below average summer or winter temperatures. Correspondences between droughts and
high summer temperatures were found for the years 1534, 1556, 1669, 1733, 1779, 1781, and 1783. Only three
years, 1556, 1781, and 1783, were ranked as extremely warm (9). Only for 1672 there was a correspondence
between drought below average winter temperatures (7).
The low number of correspondence with the drought years for Deventer and Zutphen can indicate two aspects; 1)
drought periods did not necessarily coincide with periods of above average or extreme heat (or winter droughts
with extreme cold); 2) the series of temperatures provided by Buisman and IJnsen do not provide precise enough
information, given the reliance on non-local sources for the reconstruction of pre-instrumental temperature records.
The first aspect is supported by studies with regard to northwestern Europe (Leijonhufvud and Retsö, 2021), which
suggest a lower influence of temperature on the severity of drought compared to precipitation. Aspect two can be
used to once again proof that the reliance on data from various distant locations is not always useful when studying
specific territories and localities. This can also be tested by using a large compiled index of drought-years for
multiple nearby territories, which is the case with the SDI.

## 7. Comparison with the SDI

The SDI was created by Camenisch and Salvisberg (2020) with the use of pre-existing precipitation reconstructions
from documentary sources for the Netherlands and Belgium, Germany, France, and Switzerland between 1315
and 1715, applying the seven-point scale index. When the data from Bern and Rouen was compared with the SDI,
only the years 1556, 1567, and 1681, were present in all three indices. The comparison between Bern and Rouen
also displayed a deviation in the data regarding certain 'megadroughts', as the extreme droughts of 1473 and 1540
were only reported in Bern. Because the SDI is based on years when a drought was reported somewhere within a
specific country, the amount of drought-years is significantly higher than in more local indices. When comparing
the data from Bern and Rouen with the SDI, the number of corresponding droughts was relatively low, namely a
total of seventeen corresponding cases out of the 87 drought-years in the SDI.
When comparing the data between 1500 and 1715 (see fig. 7), there are only eight corresponding drought-years,
out of 52 instances mentioned in the SDI for this period. These concern ten instances in total; eight specifically
with regard to Deventer (1534, 1556, 1615, 1630, 1634, 1645, 1666, and 1669), two concerning both Deventer
and Zutphen (1615 and 1669), and none specifically for Zutphen. This indicates that 44 droughts recorded in the
SDI were not found in the sources for Deventer and Zutphen, while 14 instances of drought (1588, 1589, 1597,
1612, 1629, 1633, 1638, 1650, 1652, 1662, 1667, 1672, 1690, 1696) were documented specifically for Deventer
and/or Zutphen during this period, but do not occur in the SDI. Such a rather low degree in correspondence supports



the conclusions regarding Bern and Rouen that generalised drought data cannot easily be applied to reconstruct or
strengthen knowledge of the specific local droughts. In fact, it shows that local sources can provide insights into
droughts that may not appear in compiled data-sets, which prompts the need to do more in-depth research for
multiple regions and localities to minimise faulty generalisations about the widespread effects of drought on
different parts of society.

**8. Comparison with the OWDA**
Camenisch and Salvisberg (2020) also compared their findings with the OWDA, a freely accessible online
database that provides year-by-year data – either via a dataset or an interactive map – of drought severity
throughout Europe and certain parts of North Africa and the Middle East on a 0.5 degrees latitude/longitude grid,
going back as far as 0 CE and coming to a halt in 2012. The OWDA displays drought-severity on a scPDSI scale
from extremely dry (-6) to extremely wet (6). It is based on a vast amount of dendrochronological data for Europe,
completed with additional information historical data on hydroclimatic extremes, but only with regard to spring
and summer drought conditions (Cook et. al., 2015). This is also the main setback of the OWDA, as it can only be
used to compare drought conditions from June to August. Another pitfall is the scPDSI ranking-system, which has
to be calibrated to other forms of indices, such as the seven-point Pfister index or the HSDS. Camenisch and
Salvisberg tested the OWDA against the data from individual indices of Bern and Rouen, as well as the SDI. They
used the censure of -2.5 on the scPDSI scale as the mark of moderate to severe and extreme droughts. As expected,
the comparison with the wider SDI yielded the most results that can be regarded as statistically significant using
the Pearson correlation ($r = 0.42$).
For the comparison with the HSDS for Deventer and Zutphen, grid snapshots were generated for each
reconstructed drought year, using the area which includes Deventer and Zutphen (52.34 to 52.ºN, and 6 to 6.48
ºE) (see figure 8). Only values of -2.5 or lower were taken into account, and no usable data was available for the
years 1638 and 1662. The outcome of the comparison was rather meagre, as from eleven drought years
corresponding to relevant outcomes of the OWDA survey (1534, 1615, 1630, 1634, 1652, 1666, 1669, 1753, 1790,
1793, and 1794), only one year, 1666, was relevant as it fell within the range of summer (JJA) drought. Another
interesting aspect is that some of the major summer drought-years, such as 1783, only receive a ranking of -2 on
the scPDSI scale of the OWDA. However, the OWDA data for certain years, such as 1615, 1630, 1669, and 1793,
which indicate autumn and winter droughts, could perhaps indicate that the effects of the summer droughts was
still felt during the following seasons. Perhaps the reconstructions using the OWDA are susceptible to the same
criticism as the comparisons to the Buisman-IJnsen series and the SDI. They strongly deviate from the drought
years reconstructed for Deventer and Zutphen, which indicates the more localised character of most droughts. Yet
it also shows the limits of dendroclimatological analysis on the basis of tree rings as a proxy for drought, which
highlights the value of using documentary sources as a means to verify the occurrence of historic droughts (Bothe
et. al., 2019; Pribyl, 2020).




**9. Discussion and Conclusion**

This article aimed to provide the first documentary evidence-based look at pre-instrumental droughts in the eastern Netherlands between 1500 and 1795, focusing on two case studies: the cities of Deventer and Zutphen. This was done by 1) examining the possibility of urban municipal archives to reconstruct past droughts; 2) creating drought indices for both cities; and 3) by comparing the gathered data with other indices to spot possible correspondence.

The archives of Deventer and Zutphen contain plenty of municipal records that provided impact-based instances of drought from the early sixteenth to the late eighteenth century. For Deventer, slightly longer-running and a larger amount records are available compared to Zutphen, where consistent records, such as daily resolutions date back from the second half of the sixteenth century. Nevertheless, similar examples of drought-related measures were found that indicate how droughts affected both cities primarily in terms of hydrological circumstances. The most common issues are related to low water levels in the rivers and canals around the city hampering navigation and low groundwater tables leading to a lack of water in wells and pumps. The main problem with the information from the documentary evidence from both archives is that although it provides a good view on the impact of drought in cities like Deventer and Zutphen, it remains difficult to establish the exact duration of droughts. The extent of droughts is only mentioned in terms of general wordings like 'prolonged' and 'extraordinary. As of such, the seven-point index, in which drought-severity is measured according to monthly thresholds, cannot be applied the data found for Deventer and Zutphen.

The alternative, creating and index along the HSDS, applies better to the source-material, but is less precise as the seven-point index, which is also calibrated using an instrumental reference-period. Nevertheless, using the HSDS for Deventer and Zutphen has led to an index with 33 droughts of varying severity on the scale of 1 (deficiency of precipitation) to 5 (widespread societal crisis) for the period 1500–1795. As is the case with municipal records, only extreme instances of drought are reported, most of which appeared to fall within the range of scale 3 and 4, denoting primarily hydrological droughts in the forms of dried up waterways, wells, and pumps. Widespread societal disruption in terms of scale 5 was not discovered in the sources, which indicates that none of the droughts had a disturbing rather than a crippling effect on society. The data from both cities also suggests a difference in seasonality, as there seems to be an unequal distribution between spring and summer droughts. There were also notable differences between similar indexed drought years for both cities, by which the effects of drought were reported differently to indicate similar levels of severity, for example by referring to dried up wells in Deventer and shut-down watermills in Zutphen. Although both instances indicate a scale 4 drought on the HSDS, referring to hydrological circumstances leading to socio-economic drought, it can be questioned whether both examples were considered as equally severe by contemporaries; was a low-water mark in wells and pumps considered just as bad as a period without the ability to employ watermills? The descriptive nature of the HSDS makes it a valuable index for the study of qualitative data from municipal records, although the next step should be to calibrate such data according to a more precise scale. Such a scale should be based on different conceptions from contemporary records to determine drought-severity more precisely. This can be done by extending the categories into different levels of, for example, hydrological drought. For instance, a lack of navigation and lay-off of watermills can be regarded as more critical or disastrous compared to a general shortage of water for domestic purposes like cooking and washing, while the need for a stable availability of water for firefighting purposes could be regarded as more important regarding the wide-ranging socio-economic effects a major fire could have on the city as a whole



(Garrioch, 2018). A next step to  in creating a more specified index for descriptive drought data that follows even
more strictly the perception of drought by contemporaries, instead of generalised criteria.
Comparison with other indices, such as the Buisman-IJnsen temperature series, the SDI, and the OWDA, have
yielded different insights with regard to the data from this study. The comparison with Buisman-IJnsen turned out
to be unfruitful, probably because temperature was of less influence on these droughts, and because the data from
multiple areas outside of the Netherlands cannot be used to create regional or local reconstructions of extreme
temperatures. The comparison with the SDI for the sixteenth and seventeenth centuries led to a limited number of
corresponding drought years, which indicates that such supra-regional indices do not correspond one-on-one with
more localised documentary-based drought reconstructions. The same can be said of the comparison with the data
gathered from single-year based snapshots from the OWDA. In this case the correspondence was even lower
regarding the sole focus on summer droughts, although the indications for certain years could point towards
possible long-lasting effects of summer droughts during consequent months.
All in all, the data for Deventer and Zutphen displays both evidence for a small number wider supra-regional
droughts as well as a larger number of local droughts specifically mentioned in the documentary sources for the
period under study. These concern primarily moderate to severe instances of drought that impacted society and
prompted responses from the city government to avert possible negative outcomes, such as food and water
shortages. As such, the source material to reconstruct droughts is closely connected to the societal responses to
drought, which indicates that specific instances of drought, primarily hydrological drought, impacted society not
necessarily by causing a widespread crisis but by limiting the use of water and waterways. The urban sources also
record very little instances of agricultural drought, of which only once instance was found for a 300-year period.
Remarkable is also that, at least for Deventer, the 'megadrought' of 1540 is entirely absent in the sources. As
Camenisch and Salvisberg (2020) demonstrated, however, this is not rare with regard to more localised
reconstructions. Although major European drought events as in 1540 feature widely in supra-regional indices,
which are comprised of documentary and natural proxy data from across different regions (Wetter e.t. al., 2014),
they are less likely to show in more local, urban analyses. Drought reconstructions for specific locations, whether
cities or villages with adequate data density, therefore should be taken into account when compiling large-scale
drought reconstructions, to gain a more accurate picture of the regional and local spread of drought and its severity
in terms of societal impact.
However, comparisons between specific, localities is another aspect that requires more attention. Deventer and
Zutphen, for example, despite their similarities and close proximity to one another yield a number of different
drought years. This can be explained, in part, by a difference in source-density for specific periods. More and
longer-running series of sources were available for Deventer, but considering the relative consistency and duration
of the municipal records for both cities it could also be argued that droughts were not always perceived as equally
menacing. Explanations for this can be found in the source-type, municipal records, which mostly refer only to
high-impact drought-events that required a governmental response, but also at the local level, for example by
studying the hydrological, geological, and socio-economic aspects of each city. This would include the dependence
of specific water sources for a city's economy, such as the need to operate watermills, or the general system of
water provisioning and how this was impacted across different areas within a city. Differing hydrological or socio-
political means that strengthened or helped to alleviate the effects of past drought could thus play an important



part in determining the severity of drought on a local level (Metger and Jacob Rousseau, 2020). This could provide
a better image of droughts through human actions and natural circumstances that have an influence on the local
impact and severity of drought and other climatic hazards, which counts not only for the past but also the future
(Degroot et. al., 2021; Kchouk et. al., 2021; Savelli et. al., 2022; Van Loon et. al., 2016). Further research is
needed, however, to draw broader conclusions on the specific local impacts of urban droughts and how this was
influenced by local natural or human factors.

**Data availability**
The data used in this article is included in the supplement and supported by appendix 1. The archival sources used
for the research of this paper are publicly and/or digitally accessible via the websites of the HCO
(https://collectieoverijssel.nl/) and ZuRAZ. (https://erfgoedcentrumzutphen.nl/) and can be found in appendix 2.
The SDI is available as a supplement to the article by Camenisch and Salvisberg (https://doi.org/10.5194/cp-16-
2173-2020). The OWDA can be freely consulted via the project website (http://drought.memphis.edu/OWDA/).
**Supplement**
The supplement related to this article is available via: https://doi.org/10.17026/dans-x3p-camy
**Competing Interests**
The authors declare that they have no conflict of interest.
**Acknowledgments**
This article was written as part of the research project: Coping with drought. An environmental history of drinking
water and climate adaptation in the Netherlands. Funding and necessary support for this research was provided by
the Dutch Research Council (Nederlandse Organisatie voor Wetenschappelijk Onderzoek, NWO).
**Financial support**
This research has been fully supported by Dutch Research Council (NWO) under file number 406.18.HW.015.











*Appendix 1: Overview of drought events from Deventer (D) and Zutphen (Z), 1500-1795*

| Year | Month | Season | Location | HSDS ranking | Source | Descriptions from the sources |
|---|---|---|---|---|---|---|
| **1534** | Uncertain | Summer | Deventer | 3 | HCO (0690, 135;1) | Great drought and a lack of water, ordinance calls for storing water in barrels in case of a fire |
| **1556** | September | Autumn | Deventer | 3 | HCO (0690, 135;3) | Period of drought and a lack of water, calls for storing water in barrels in case of a fire |
| **1588** | January | Winter | Zutphen | 4 | ZuRAZ (0001;2) | Period of drought and use of the watermills only permitted after rainfall returns |
| **1589** | December | Winter | Zutphen | 3 | ZuRAZ (0001;2) | Low water levels, no navigation possible to the city |
| **1597** | March | Autumn | Zutphen | 2 | ZuRAZ (0001;3) | low water levels, limited navigation |
| **1612** | July | Summer | Deventer | 3 | HCO (0691;7a) | Period of drought and low water levels on the IJssel, a lack of water in wells and an ordinance calls for storing water in barrels in case of a fire |
| **1615** | September (D) and May (Z) | Autumn (D) and Spring (Z) | Deventer and Zutphen | 3 (D) and 2 (Z) | HCO (0691;6a); ZuRAZ (0001;6) | Period of drought, request from the sworn men in Deventer to issue a an ordinance requiring the inhabitants to store water |
| **1629** | August | Summer | Deventer and Zutphen | 3 | HCO (0691; 7b)ZuRAZ (0001;8) | Period of drought and low water, limited to no use of watermills (and windmills) and increased use of draught mills in Zutphen, an ordinance calls for storing water in barrels in case of a fire in Deventer |
| **1630** | December | Winter | Deventer | 2 | HCO (0691; 6b) | Period of low water, request from the sworn men to construct a palisade as extra protection of the city due to the low water mark in the IJssel |
| **1633** | September | Autumn | Deventer | 3 | HCO (0691; 7b) | Period of major drought, low water marks and an ordinance calls for storing water in barrels in case of a fire |
| **1634** | March | Spring | Deventer | 4 | HCO (0691; 6b) | Period of drought, low water addressed by the sworn men to deepen the city's harbour, lack of wind and water necessitates to prepare the draught mill |



| | | | | | | |
|---|---|---|---|---|---|---|
| **1638** | August | Summer | Deventer | 3 | HCO (0691; 7b) | Period of drought, low water levels and a lack of water in wells, an ordinance calls for storing water in barrels in case of a fire in Deventer |
| **1645** | August | Summer | Deventer | 3 | HCO (0691; 7b) | Period of drought, lack of water in wells, an ordinance calls for storing water in barrels in case of a fire |
| **1650** | June | Summer | Deventer | 4 | HCO (0691; 14.4) | Period of drought, lack of water, watermills unable to function |
| **1652** | May | Spring | Deventer | 3 | HCO (0691; 7b/14.4) | Period of drought, lack of water, ordinance calls for storing water in barrels in case of a fire, use of water from a nearby brook to wet the St. Jurriensdijk instead of water from the dried out Schipbeek |
| **1662** | October | Autumn | Deventer | 3 | HCO (0691; 7b) | Lack of water in the city's wells, ban on lighting fireworks and an ordinance calls for storing water in barrels in case of a fire |
| **1666** | July | Summer | Deventer | 3 | HCO (0691; 7c) | Period of drought and an ordinance calls for storing water in barrels in case of a fire |
| **1667** | August | Summer | Zutphen | 3 | ZuRAZ (0001;18) | Lack of water in the city, fear for fires, ban on lighting fireworks or celebratory pitch barrels |
| **1669** | September | Autumn | Deventer and Zutphen | 4 (D) and 3 (Z) | HCO (0691; 7c); ZuRAZ (0001;18) | Extraordinary drought, unusually low water levels and a lack of water in the wells of Deventer leading to an enforced deepening of wells, an ordinance calls for storing water in barrels in case of a fire in Zutphen |
| **1672** | January | Winter | Deventer | 3 | HCO (0691; 7c) | Low water levels in the rivers, a lack of water in wells, storing water in barrels |
| **1690** | May | Spring | Deventer | 3 | HCO (0691; 7c) | Period of drought, ordinance calls for storing water in barrels in case of a fire |
| **1696** | December | Winter | Deventer | 3 | HCO (0691; 7c) | Lack of water in the city, ban on certain activities causing fire hazards |
| **1731** | November | Winter | Deventer | 3 | HCO (0691; 7d) | Waterless period, general lack of water in the city, an ordinance calls for storing water in barrels in case of a fire |



| 1733 | uncertain (D) September (Z) | Summer (D) autumn | Deventer and Zutphen | 3 (D) 4 (Z) | HCO (0691; 4.28) ZuRAZ (0001;32) | Dry summer, low water levels and a lack of water in wells and pumps in Zutphen, deepening of the waterways near Deventer |
|---|---|---|---|---|---|---|
| 1749 | October andDecember | autumn/winter | Zutphen | 3 | ZuRAZ (0001;35) | Extraordinary low water levels on the IJssel river, a lack of peat due to hampered navigation |
| 1753 | June (D) and September (Z) | Summer (D) and autumn (Z) | Deventer and Zutphen | 4 | HCO (0691; 7e) ZuRAZ (0001;37) | Excessive drought, low water levels, many wells without water, an ordinance calls for storing water in barrels in case of a fire |
| 1772 | December (D) and October (Z) | Autumn (D) and winter (Z) | Deventer and Zutphen | 3 | HCO (0691; 7f;)ZuRAZ (0001;46) | Long-lasting lack of water in the rivers, lack of water in wells and pumps, requests to limit water use |
| 1779 | April (D) and February (Z) | Winter (D) and spring (Z) | Deventer and Zutphen | 4 | HCO (0691; 7f); ZuRAZ (0001;49) | Long-lasting lack of water in the rivers, many wells without water, ban on certain water-using activities, no navigation possible to Zutphen, outbreaks of dysentery in Zutphen |
| 1781 | March (D) and February (Z) | Winter (D) and spring (Z) | Deventer and Zutphen | 4 | HCO (0691; 4.43); ZuRAZ (0001;50) | Long-lasting drought and low water levels in the rivers, no navigation and watermills out of use specifically in Zutphen |
| 1783 | August (D) and July/August (Z) | Summer | Deventer and Zutphen | 4 | HCO (0691; 7f); ZuRAZ (0001;52) | Excessive drought, lack of water in the rivers, many wells without water, watermills out of use, limited yields of buckwheat near Deventer as a result of drought, an ordinance calls for storing water in barrels in case of a fire and outbreaks of dysentery in both cities |
| 1790 | April | Spring | Deventer | 3 | HCO (0691; 7g) | Strong drought, many wells without water, ordinance to limit water use. |
| 1793 | October | Autumn | Zutphen | 3 | ZuRAZ (0001.122) | Very low water levels, lack of water, ban on using water from the communal wells for scrubbing of streets |
| 1794 | January | Winter | Zutphen | 3 | ZuRAZ (0001.122) | Very low water levels, lack of water, ban on using water from the communal wells for scrubbing of streets |







**Appendix 2: Archival sources**

Historisch Centrum Overijssel (HCO) (Regional Archives of Overijssel), Deventer, Stad Deventer, periode Middeleeuwen, 1241-1591 (ID 0690), Edicta magistratus die buyrspraecht genoemptt or Dat boick der buyrspraiken, 1459-1538, 1555-1596, 135.1, 3.

Historisch Centrum Overijssel, Deventer, Schepenen en Raad van de stad Deventer, periode Republiek 1591-1795 (ID 0691), Prothocoll des Rades van dagelicken resolutien, or Liber quotidianarum resolutionum civitatis Daventriensis, 1591-1795, 4.14,

Historisch Centrum Overijssel, Deventer, Schepenen en Raad van de stad Deventer, periode Republiek 1591-1795 (ID 0691), Register van resolutien van Schepenen en Raad en Gezworen Gemeente (Concordaten), 1600-1794, 6a-m.

Historisch Centrum Overijssel, Deventer, Schepenen en Raad van de stad Deventer, periode Republiek 1591-1795 (ID 0691), Register van verordeningen en bekendmakingen van het stedelijk bestuur (Buurspraakboek) or Liber publicationum, 7a-g.

Erfgoed Centrum Zutphen (ZuRAZ) (Regional Archives of Zutphen and surrounding areas), Zutphen, Oud-Archief van de stad Zutphen, 1206-1815 (ID 0001), Memorien- en resolutieboek van de stad Zutphen, registers van resoluties van de magistraat, 1573-1808, 2, 3, 6, 8, 18, 32, 35, 37, 46, 49, 50, 52.

Erfgoed Centrum Zutphen, Zutphen, Oud-Archief van de stad Zutphen, 1206-1815 (ID 0001), Repertoria op de resoluties van de magistraat, 1573-1620, 110.

Erfgoed Centrum Zutphen, Zutphen, Oud-Archief van de stad Zutphen, 1206-1815 (ID 0001), Repertoria op de resoluties van de magistraat, 1620-1660, 111.

Erfgoed Centrum Zutphen, Zutphen, Oud-Archief van de stad Zutphen, 1206-1815 (ID 0001), Repertoria op de resoluties van de magistraat, 1661-1700, 112.

Erfgoed Centrum Zutphen, Zutphen, Oud-Archief van de stad Zutphen, 1206-1815 (ID 0001), Repertoria op de resoluties van de magistraat, 1701-1740, 113.

Erfgoed Centrum Zutphen, Zutphen, Oud-Archief van de stad Zutphen, 1206-1815 (ID 0001), Repertoria op de resoluties van de magistraat, 1741-1780, 114.

Erfgoed Centrum Zutphen, Zutphen, Oud-Archief van de stad Zutphen, 1206-1815 (ID 0001), Repertoria op de resoluties van de magistraat, 122.

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
