# Peer review of "Documentary evidence of urban droughts and their impact in the"

_EGUsphere, 2022_

## Author Response (AR3)

Reactions to referee comments and changes for the 4th submission

(New additions are marked in red)

Editor's comments:

*Thank you for the revised manuscript and thanks as well to the reviewers for a thorough job. I would like to ask for a couple of minor revisions around the additional analysis paragraph.*

*"When looking at the annual average temperature (in ∘C) for the summer months (JJA), a statistical comparison shows a*
*rather weak Pearson correlation (r=0.17) for the Deventer and moderate correlation (r=0.45) for Zutphen. This*
*suggest very weak to moderate correlations between the annual average summer temperatures and the HSDS for*
*either city. Comparing the average annual temperature series with the HSDS led to an even weaker (r=0.04)*
*correlation for Deventer, and a moderate correlation (r=0.52) for Zutphen. However, it must be noted that due to*
*the small set of years, these results only bear a rather low level of statistical significance."*

*Please add in significance values - what for example is the significance value for P for the two correlations of r=0.45 and r=0.52? I suspect (please also state n) neither are statistically significant at 95%? I would also ask for a rephrasing of the 'moderate' correlation. An r value of 0.45 means 20% of the variance in the series is explained, that's not really moderate.*

*So minor edits - i'm going to ask the authors to please review the language around how strong correlations are, add in significance at P whilst also stating n (it's possible to get statistically significant values at p with very low r if n is high enough etc ) . Please also review your conclusions based on the above.*

Reaction:

I have taken into account the comments from the editor with regard to the comparisons, which can be found under comment 3 for referee 1 (to which the request for cross-comparison belonged). For the rest, I have looked through the text again and performed some minor editing where necessary, usually considering typos.

RF 1:

Major comments:

1. *The topic of "urban drought" may be new to many readers, and this needs to be better introduced. Is it really an "urban drought" (having impacts within the city walls, as it is said at one instance) or is it "urban information" on drought (or "urban responses" to drought - would the city respond to a rural drought)? What do these indices capture? What is actually relevant, and why? This might also relevant for future climate (an aspect which is a bit missing). Just a few more word on that would be appreciated.*

I have added a more elaborate explanation of 'urban drought' in the introduction (L.73-79), where I explain that it concerns the impact of droughts on cities, taken from urban documents regarding the impacts and responses. This is also the main quality of the HSDS, which captures drought severity in terms of the impact and responses by the urban government/populace.

2. *Generally I am surprised by the large differences between the two cities in all aspects of droughts (almost no coincidence of drought years, different seasonality etc.). This seems to be an important finding that should be better discussed. For that, it would be good to show a map with the two cities and the hydrogeography. Are they in the same river catchment? Is the land use surrounding the two cities similar etc.*

I have added additional information to the section regarding the 'outcomes' primarily with regard to specific geohydrological qualities, in particular the different streams that provide water to the cities. A customised hydrographic map has been added (p.9) to provide additional information about these rivers, also with regard to the location of both cities within the eastern Netherlands.

3. *The index method, HSDS etc. is well ecplained including a critical discussion. Nevertheless, as reader I would like to know whether (in the literature) the HSDS concept has been cross-validated or cross-compared. This might also be relevant for the discussion of the comparison with other data sets (and methofds) in this paper. Just thinking about the evidence in this paper, I could imagine that, e.g., the way in which navigability is affected may vary a lot from river to river, so this part of the definition will capture differences between cities).*

Together with a colleague I have reviewed the correlations from the previous submission and made the decision to rerun these using the Spearman correlation coefficient instead of the Pearson test, given the fact that the data for the HSDS for Deventer and Zutphen are not normally distributed. These results have replaced the Pearson correlations from the previous version, also noting the n and p-values for each case. The latter is included in the conclusions regarding the strength of the correlations and significance of the analysis

4. *L. 199: Only nine coinciding years seem to be very little, but given the low number of drought years in total it is arguably still highly significant. Here I would expect more information: Are the most severe droghts in one city at least HSDS-2 events in the other one? Just some more quantitative evaluation would be interesting. How likely is it that two cities with same meteorological drought (probably this is almost always the case for the two cities) have different hydrological drought? (Again, questions of that sort might be relevant also for future climate).*

Further clarification with regard to this aspect has been added in the paragraph regarding the outcomes (L. 244-255). This comes back to the differing geohydrological situations that were addressed in the previous comment, which explain why there could be differences in drought severity for coinciding years.

5. *Comparisons: I appreciate the three comparisons performed, although the results are perhaps a bit disappointing at first. It would also be interesting to look at the mutual comparisons of these three data sets.*

I have now included quantitative comparisons between the HSDS for Deventer and Zutphen with regard to the other datasets. Pearson correlations were generated with mixed results regarding the similarities between the datasets. Given the small set of years for the HSDS, the level of statistical significance of these analyses remains low. Yet this is also a relevant outcome, which prompts more additional research to extend the dataset and strengthen the HSDS.

6. *Overall the paper is relaively long. When revising the paper, please observe length.*

I understand that the paper might be perceived as a bit lengthy, but I believe that it remains well within the scope of a research article discussing such a topic. Nevertheless, I have cut some words with regard to specific details, i.e. about dysentery etc. (p. 11, L. 304-310). Within the 'data' paragraph (p.3-4), I have also shortened the information with regard to the sources. In the part discussing specific years, I have cut the paragraphs regarding two years and left only the three most significant years. I have also cut appendix 1 entirely and now aim to provide it as a separate, digitally available data sheet.

Minor comments:

*The Figures seem straight from Excel. Please draw them more neatly and with better x-axes. This is sometimes hard to understand (narrow lines, odd years, no tickmarks, etc.). Brush up the figures.*

I have done this accordingly to make the figure more understandable and neat.

*L. 49: "This is a trend.." What trend (unclear)?*

In hindsight the use of the word 'trend' is inappropriate and unnecessary, as it refers to the general pattern of precipitation (deficit). Therefore, I have removed the word 'trend' in this case and replaced it with more clear wording.

*L. 220: agricultural drought (only 1): Is this also within the city limits or are these reports referring to the rural surroundings?*

I have added additional information regarding this aspect. which explains that it concerns agricultural activity in the cities' hinterlands that is reflected in the market prices of certain foodstuffs (p. 7, L. 233-236).

*Fig. 2, Zutphen: Looks visually inhomogeneous. Is this due to the source density? (if so, perhaps mark in the figure) (?)*

This has to do with the difference in source density between Deventer and Zutphen. In the first graphs I used the drought years of Deventer and Zutphen combined in both graphs, which led to the inhomogeneous picture. I have altered this by redrawing both graphs for Deventer and Zutpen, using only drought years pertaining to each city, which provides a more clear image of the distribution of different drought-types

*Fig. 2: Are these droughts per decade? This is not clear.*

It concerns individual drought years. This is now better explained as part of the sub-text for fig. 2.

*Fig. 3: The difference in seasonality is just huge!*

This has been further explained in the 'outcomes' paragraph as part of the differences in source density between both cities. (p. 6-10).

*L. 255 (and others): Do not start title with a number*

I have changed these titles (L.255, 266, 275, 284, 292) to 'The year 1669' etc.

*van Loon et al. 2016 is not in the reference list*

This has been added to the list of references.

*Vörösmarty et. al., 2004 is not in the reference list (there might be more; I have not checked systematically)*

This has been added to the list of references.

RF 2

Major comments:

*I'm sitting on an island without computer and limited IT-structure, so sorry for shorthanded comments. I think this is an interesting paper that would be even better if more structured.*

*This is an interesting paper showing local resilience to drought in 2 dutch cities during early modern times. Main critic is that it is rather unclear if the study aims at climatological research or something else. IF the author wants to pinpoint climatological impact on society, I think he has some more precise exercise to do.*

*I am not familiar with the Netherlands - and not at all w the local area of deventer and zutphen. My first thought to the result of more droughts in 2nd half of 18th century (row 244-245) was that maybe there had been major drainage of the land in the late 17th and early 18th century?*

The area around Deventer and Zutphen does not belong to the parts of the Netherlands where major drainage projects had taken place during the early modern period. However, the differing geohydrological conditions are now addressed in the revised version as part of the 'outcomes' paragraph.

*I think the author succedes with his 3 aims. (Row 16 - 20)*

1. *49-51. Unclear sentense. Even substituting 'trend' to 'fact'. The subordinate clause cause confusion. OR the sentence is out of context.*

This has been altered, also with regard to the comments made by RF 1.

2. *90-94. The author wants to prove that data from the Netherlands are good, because Spanish data are. That may be so, since NL was subject to Spain during part of the period, but it is not obvious*

This has been clarified in the revised version.

3. *93. Reference Escayol & Barriendos, 2021. Is not in reference list.*

This has been fixed.

4. *109 "...daily resolutions and can be...": delete "and".*

This has been fixed.

5. *109-110. "This series... provided..." THESE series. And maybe present tense?*

I will check again the use of past and present tense in the article and make the necessary alterations.

6.  111. "...was required". Made?

This has been altered to 'had to be carried out' to make the sentence more clear.

7.  14. New paragraph between "drought. Firstly..."

I do not believe that it is necessary to make a cut here to create an additional paragraph.

8.  114-133: maybe this should be in "Methodology"?

I have moved these lines to the end of the 'methodology' section.

9.  158-185. Move into "Data"? Or make a new chapter "theory/previous research"?

I do not believe it is necessary to create an entirely new chapter with regard to the information in these lines, as it concerns an explanation of the applied methodology based on previous studies with a similar approach/method.

10. I think the text would improve if 207-214 salvisberg, 2020; garnier, 2019) was moved to 195: just below "4. Outcomes" or put into "Theory".

This part has been moved just to the start of the 'outcomes' paragraph.

11. 199. ...coinciding years. Hydrological droughts... New paragraph Hydrological...

I can see why a cut to a new paragraph would make sense here, but I do not believe it is highly necessary to do so. Therefore, I have dismissed this comment.

12. Figure 1: Question! No year has insufficient data? (This is just a question. No criticism.)

This has been explained revised version in the 'outcomes' section. The quick answer is that none of the data I found falls within this category, hence the absence.

13. 228-232: move to above 224. (Then all text concerning figures come before the figure).

This has been fixed.

14. 254 "...has been restricted the most..." -> "...has been restricted to the most..."

This has been fixed.

15. 257 "...and pumps the" -> "... and pumps in the"

This has been fixed.

16. 260. New paragraph "For Zutphen, references..." ?

I do not believe it is necessary to start a new paragraph here, given its already compact size.

17. 270. New paragraph "In Zutphen, the 1733..." ?

I do not believe it is necessary to start a new paragraph here.

18. 295 "...mentioned as the great spring drought led to..." -> "...mentioned as "the great spring drought", which led to..."

This has been fixed.

19. 300-304: too much about dysentery. Kill your darlings!

This has been done, also with regard to the comment by RF 1.

20. 307. "likely the main drivers..." -> "likely to be the main drivers..."

This has been fixed.

21. 350. "... between drought below average..." -> "... between drought and below average..."

Agreed. This will be altered in the revised version.

22. 351-354: what do modern data show when comparing drought to temperature (for zutphen & deventer)? Maybe you should include such a comparison in this paper? I.e. compare drought today w british temperature.

This has been addressed in the introduction as part of the explanation of the region of interest. I have made a more detailed remark on the differences between the western coastal and eastern inland provinces of the Netherlands, for which a recent study has shown differences in precipitation deficit and temperature.

23. 378-382. Very good

24. 400. Where is figure 8?

Thanks for pointing this out. It concerns a figure that I decided not to add to the article right before submission. The reference has been removed in the revised version.

25. 439. "[None] ...had a disturbing rather than a crippling effect..." -> "[none]..had a crippling effect but  rather a disturbing effect.."  [this is not very good. But certainly NONE of the droughts had a crippling effect...]

This has been fixed.

26. 454-455. "A next step..." this sentence seem out of context. Write some more!

This sentence has been removed as it was deemed unnecessary.

27. 457-464: an awful lot of "the". I think text'd be better if cutting out most of the "the:s"

I have looked into this and made changes.

28. 465 "...during consequent months" -> "... for following months". (Or possibly "consecutive")

This has been fixed.

29. 466. "...data /-/ displays.." -> "...data /-/ display.."

Agreed. This will be changed accordingly in the revised version.

30. 525 appendix

This comment is unclear to me.

31. 1615... "deventer to issue a an ordinance" -> "deventer to issue an ordinance"

This has been fixed.